# The Translocator Protein (*TSPO*) Genetic Polymorphism A147T Is Associated with Worse Survival in Male Glioblastoma Patients

**DOI:** 10.3390/cancers13184525

**Published:** 2021-09-08

**Authors:** Katie M. Troike, Arlet M. Acanda de la Rocha, Tyler J. Alban, Matthew M. Grabowski, Balint Otvos, Gino Cioffi, Kristin A. Waite, Jill S. Barnholtz Sloan, Justin D. Lathia, Tomás R. Guilarte, Diana J. Azzam

**Affiliations:** 1Department of Cardiovascular & Metabolic Sciences, Lerner Research Institute, Cleveland Clinic, Cleveland, OH 44195, USA; troikek@ccf.org (K.M.T.); albant@ccf.org (T.J.A.); grabowm2@ccf.org (M.M.G.); otvosb@ccf.org (B.O.); lathiaj@ccf.org (J.D.L.); 2Department of Molecular Medicine, Cleveland Clinic Lerner College of Medicine, Case Western Reserve University, Cleveland, OH 44195, USA; 3Department of Environmental Health Sciences, Robert Stempel College of Public Health & Social Work, Florida International University, Miami, FL 33199, USA; aacandad@fiu.edu; 4Department of Neurosurgery, Cleveland Clinic, Cleveland, OH 44195, USA; 5National Cancer Institute, Division of Cancer Epidemiology and Genetics, Trans-Divisional Research Program, Bethesda, MD 20892, USA; gino.cioffi@nih.gov (G.C.); kristin.waite@nih.gov (K.A.W.); jill.barnholtz-sloan@nih.gov (J.S.B.S.); 6National Cancer Institute, Center for Biomedical Informatics and Information Technology, Bethesda, MD 20892, USA; 7Case Comprehensive Cancer Center, School of Medicine, Case Western Reserve University, Cleveland, OH 44195, USA; 8Rose Ella Burkhardt Brain Tumor and Neuro-Oncology Center, Cleveland Clinic, Cleveland, OH 44195, USA; 9Brain, Behavior & the Environment Program, Robert Stempel College of Public Health & Social Work, Florida International University, Miami, FL 33199, USA

**Keywords:** *TSPO*, biomarker, glioblastoma, single nucleotide polymorphism, survival

## Abstract

**Simple Summary:**

The translocator protein 18 kDa *(TSPO)* gene is highly expressed in glioblastoma (GBM), the most common primary malignant brain tumor, which remains one of the most difficult tumors to treat. *TSPO* is located in the outer mitochondrial membrane and binds cholesterol through its C-terminal domain. One frequent single-nucleotide polymorphism (SNP) rs6971, which changes the alanine 147 into threonine (Ala147Thr), has been found in the C-terminal domain of the *TSPO* region and dramatically alters the affinity with which *TSPO* binds drug ligands. However, the potential association between the *TSPO* genetic variants and GBM clinical outcomes is not known. Here, we evaluated the effects of the Ala147Thr SNP localized in this *TSPO* region on biological, sex-specific, overall, and progression-free GBM survival. Our findings suggest an association between the *TSPO* rs6971 variant and adverse outcomes in male GBM patients but not in females. These findings also suggest that the *TSPO* rs6971 SNP could be used as a prognostic marker of survival in GBM patients.

**Abstract:**

Glioblastoma (GBM) is the most common primary brain tumor in adults, with few available therapies and a five-year survival rate of 7.2%. Hence, strategies for improving GBM prognosis are urgently needed. The translocator protein 18kDa *(**TSPO*) plays crucial roles in essential mitochondria-based physiological processes and is a validated biomarker of neuroinflammation, which is implicated in GBM progression. The *TSPO* gene has a germline single nucleotide polymorphism, rs6971, which is the most common SNP in the Caucasian population. High *TSPO* gene expression is associated with reduced survival in GBM patients; however, the relation between the most frequent *TSPO* genetic variant and GBM pathogenesis is not known. The present study retrospectively analyzed the correlation of the *TSPO* polymorphic variant rs6971 with overall and progression-free survival in GBM patients using three independent cohorts. *TSPO* rs6971 polymorphism was significantly associated with shorter overall survival and progression-free survival in male GBM patients but not in females in one large cohort of 441 patients. We observed similar trends in two other independent cohorts. These observations suggest that the *TSPO* rs6971 polymorphism could be a significant predictor of poor prognosis in GBM, with a potential for use as a prognosis biomarker in GBM patients. These results reveal for the first time a biological sex-specific relation between rs6971 *TSPO* polymorphism and GBM.

## 1. Introduction

Glioblastoma (GBM) is the most common primary malignant brain tumor in adults and accounts for 14.5% of all primary brain and central nervous system (CNS) neoplasms [1]. In the U.S., the incidence rate for GBM is 3.23 per 100,000 in the population, the highest for malignant tumors, and is 1.6 times higher in males than females [1]. Despite the aggressive and multimodal standard-of-care treatment, which includes maximal safe surgical resection followed by radiation in addition to concomitant and adjuvant chemotherapy, the prognosis remains extremely poor, with a five-year survival rate of 7.2% [1]. The median survival also shows a biological sex-based disproportion of 20.4 months for females and 17.5 months for males [2]. 

The unfavorable prognosis for GBM can be attributed to numerous features, including the intra- and inter-heterogeneity and intrinsic cell plasticity [3], a high rate of invasion to the brain parenchyma, a hypoxic intratumoral environment [4], the presence of cancer stem cells that contribute to the treatment resistance and recurrence [5], and an immunosuppressive tumor state [6]. Moreover, increasing evidence suggests that mitochondrial dysfunction plays a key role in the pathogenic events of GBM due to their role as central regulators of cell metabolism, cell death, oxidative stress, invasion, and inflammation [7]. 

The translocator protein 18 kDa (*TSPO*), previously known as the peripheral benzodiazepine receptor [8], has been extensively studied within the last two decades due to its location in the outer mitochondrial membrane (OMM). *TSPO* is known to play crucial roles in essential mitochondria-based physiological processes, such as mitochondrial respiration, metabolism, cellular bioenergetics, cholesterol transport and steroidogenesis, and heme biosynthesis, among others [8,9,10,11,12]. *TSPO* expression is most abundant in steroid-synthesizing tissues, whereas the heart and kidney express intermediate levels of *TSPO*. Interestingly, *TSPO* is expressed at low levels in the normal brain neuropil but becomes highly expressed following nervous system insults and neuroinflammation [13,14], highlighting the role of *TSPO* as a sensitive biomarker of brain injury and neuroinflammation. Given the clinical interest, *TSPO* positron-emission tomography (PET) imaging has been used in a wide variety of neuroinflammatory conditions [15,16,17,18,19,20], including high-grade gliomas [21,22,23,24].

The human *TSPO* gene located on chromosome 22q13.3 is made up of 4 exons with a large intron, containing repetitive sequences separating the first and second exons, a GC-rich promoter region, and multiple transcription initiation sites [25,26,27]. *TSPO* is a hydrophobic protein composed of 169 amino acids with a high degree of homology among species from bacteria to humans [9]. The structure of the protein has five putative transmembrane domains in which the C-terminus is exposed to the cytoplasm [28]. The cholesterol recognition amino acid consensus sequence has been identified on the interface between the fifth transmembrane domain and the cytosol-facing C-terminus domain (L144-S159) [29]. In addition, the *TSPO* gene has several polymorphisms, but only two variants, namely rs6971 and rs6972, are the most frequent in human populations and are located in the C-terminal domain [30]. The most studied variant, rs6971, is an A/G transition in exon 4 that leads to a non-conservative amino acid substitution, Ala147Thr, in the fifth transmembrane domain of the protein. This polymorphism destabilizes the protein, especially in the ligand-binding pocket, reducing the affinity of *TSPO* for cholesterol and its transport into mitochondria [31], consequently blunting the production of steroid precursors [29,32,33] and the neurosteroid pregnenolone in lymphomonocytes [34]. This single-nucleotide polymorphism (SNP) has been associated with psychiatric disorders, such as bipolar disorder, depression, and anxiety [35,36,37,38]. The structural alteration caused by the rs6971 polymorphism also decreases the distance between the second and fifth transmembrane domains [39], which dramatically reduces the binding affinity of second-generation *TSPO*-PET radiotracers [40,41]. The second most common though less studied polymorphism, rs6972, is a G/A transition also in exon 4 of the *TSPO* gene, leading to a single amino acid substitution, Arg162His, and has been reported to affect the conformation of the C-terminus through use of structural bioinformatics models [30]. 

Several studies have shown a positive correlation between *TSPO* expression and grade of malignancy and glioma cell proliferation and a negative correlation with survival in glioma patients [42,43,44]. Noteworthy, *TSPO* expression varies significantly depending on the GBM subtype and is significantly higher in isocitrate-dehydrogenase wild-type (IDH^WT^) compared to IDH mutant (IDH^MUT^) GBM [45]. However, the association between the most frequent *TSPO* genetic variant and GBM susceptibility, prognosis, and patient outcome is not known. 

In the present study, we analyzed the association between rs6971 *TSPO* polymorphic variant and clinical outcomes for GBM patients. We used three independent cohorts of GBM patients to evaluate the correlation of *TSPO* genetic variant with the overall survival (OS) and progression-free survival (PFS) time, incorporating biological sex and treatment-specific analysis. Our results indicate allele-specific effects of *TSPO* rs6971 SNPs on the survival of GBM patients in a sex-specific manner.

## 2. Materials and Methods 

### 2.1. Study Population

The Cleveland Clinic Foundation (CCF) cohort: Peripheral blood samples from 441 GBM patients were collected through the Rose Ella Burkhardt Brain Tumor and Neuro-Oncology Center (BBTC) at the Cleveland Clinic under IRB2559. White blood cells from each blood sample were isolated via Ficoll gradient and then snap-frozen and stored at −80 Celsius for research use. In this study, we selected all available GBM samples. Genomic DNA was extracted using a Qiagen DNeasy Blood & Tissue Kit (Qiagen, Hilden, MD, USA) following the manufacturer’s protocols. DNA purity and concentration were measured using a ThermoFisher NanoDrop spectrophotometer (ThermoFisher, Waltham, MA, USA).

The Case Western Reserve University (CWRU) cohort: Newly diagnosed, untreated brain tumor patients were identified for the Ohio Brain Tumor Study (OBTS) under approval from University Hospitals IRB CC296. Clinical and pathological data were gathered for each patient. Patient blood samples were collected at the time of consent. DNA extraction from whole blood samples was conducted using a Qiasymphony robotic platform (Qiagen) with Qiagen processing kits designed to maximize DNA (QIAsymphonyDSP DNA midi kit) yields and purity.

The Cancer Genome Atlas (TCGA) cohort: Raw BAM files from the TCGA-GBM cohort were utilized for the analysis of the *TSPO* rs6971 SNP using whole-exome sequencing data aligned by the TCGA. Alignment of the SNP rs6971 genotype was identified via the use of HaplotypeCaller, where samples with alternative counts at the reference position chr22:43162920 were identified. After classification of the samples by genotype, the phenotype data were downloaded via TCGA, and survival analysis was performed. Survival analysis was performed using a log-rank test via R version 4.1.0.

### 2.2. SNP Selection and Genotyping 

Patient genotyping was performed using rhAmpTM SNP assays (IDT, custom design) for the human *TSPO* polymorphism: rs6971 (A→G). The genotyping reaction was carried out according to the manufacturer’s protocol. Briefly, 10 μL reaction volumes were prepared by combining 5.3 μL of combined master mix (IDT, 1076018) and reporter mix (IDT, 1076024), 0.5 μL rhAmpTM SNP assay, 2.2 μL nuclease-free water, and 2 μL diluted DNA sample. gBlocks gene fragments (IDT, custom design) for each SNP were used as positive controls for all genotyping. Reactions were performed in an Applied Biosystems QuantStudioTM 3 (Thermo Fisher, MA, USA) instrument with cycling parameters as specified by the assay manufacturer.

### 2.3. GlioVis Analysis

Gene expression and GBM patient survival data from the TCGA Project, the Chinese Glioma Genome Atlas (CGGA) database, REMBRANDT brain cancer dataset, and Gravendeel database, were analyzed using GlioVis (Version 0.20 (18 March 2016)) (http://gliovis.bioinfo.cnio.es/, accessed on 10 May 2021) [46]. We only included adult patients from all the databases and stratified patients by biological sex, when available. We used patients’ data from the TCGA-GBM database whose tumors had Affymetrix HG-U133A, Agilent-4502A, and RNA-seq RNA expression for our gene of interest. Pairwise comparisons using *t*-test (with Bonferroni correction) was performed. *p*-values of the pairwise comparisons are indicated in the graphs as *** *p* < 0.001; ** *p* < 0.01; * *p* < 0.05; and ns, not significant.

### 2.4. Statistical Analysis

Demographic and clinical characteristics were compared between rs6971 polymorphism genotype groups. Wilcoxon rank-sum test was used to assess differences in continuous data. Differences in categorical data were assessed through Pearson’s chi-square test and Fisher’s exact test where appropriate (cell counts less than 5 cases). Kaplan-Meier curves were generated and log-rank test performed to assess differences in OS and PFS between groups. These survival differences were also assessed for patients who had received standard of care and surgery (gross total or near total resection). Median survival and 95% confidence intervals for OS and PFS are reported. All statistical analyses were performed in R (version 4.0.5). R package gtsummary was used to generate descriptive tables, and packages survival and survminer were used to generate the survival curves. *p*-values under 0.05 are statistically significant. 

## 3. Results

### 3.1. TSPO Expression Is Upregulated in GBM Patient Samples

To evaluate *TSPO* expression in GBM, we conducted a *TSPO* mRNA expression analysis in TCGA-GBM dataset using GlioVis platform. We observed a significant increase in *TSPO* gene expression in GBM compared to that of non-tumor brain tissue (*p* < 0.001) (Figure 1A). A significant increase in the mRNA *TSPO* expression in GBM tumor samples was also observed comparing the three different RNA data platforms available from the TCGA-GBM database as well as in the REMBRANDT and Gravendeel independent datasets (*p* < 0.001) (Appendix A).

Similarly, we also found higher gene expression of *TSPO* in *IDH* wild-type GBM subtype compared to *IDH* mutant GBM subtype in the TCGA-GBM dataset (*p* < 0.001) (Figure 1B). Interestingly, no significant difference was observed when comparing females versus males in the same GBM cohort (Figure 1C). In addition, we analyzed *TSPO* expression in the three different GBM subtypes as classified by Phillips and colleagues [47,48] using the TCGA, CGGA, Rembrandt, and Gravendeel independent datasets. We found higher *TSPO* expression that was statistically significant in the mesenchymal subtype as compared to classical and proneural subtypes in the analyzed datasets (Appendix A).

A recent study reported a new pathway-based classification of GBM identifying four different subtypes that included metabolic and developmental attributes, namely proliferative/progenitor (PPR), mitochondrial (MTC), neuronal (NEU), and glycolytic/plurimetabolic (GPM). As *TSPO* is a mitochondrial protein, we analyzed its expression in these four newly described subtypes and found higher gene expression of *TSPO* that was statistically significant in the MTC subtype as compared to PPR and NEU subtypes. We also found higher *TSPO* expression in the GPM subtype as compared to the PPR subtype (Appendix A).

Additionally, we analyzed the correlation of the *TSPO* gene expression with commonly mutated genes in GBM (*PTEN, TP53, EGFR, PIK3R1, PIK3CA, NF1,* and *RB1*) using the TCGA dataset. Out of all the genes, we only found higher *TSPO* gene expression that was statistically significant in *TP53* wild-type GBM subtype compared to *TP53* mutant (Appendix A).

To investigate the association between *TSPO* mRNA expression and OS time in GBM patients, we interrogated a series of independent datasets (the Chinese glioma genome atlas (CGGA), TCGA-GBM using Affymetrix HG-U133A platform, and Gravendeel). We stratified the GBM patients in each database based on their biological sex. We found a statistically significant difference in the OS time for the entire population only in the Gravendeel dataset (*p* = 0.0024) (Figure 2). Interestingly, when GBM patients were stratified by biological sex in the three analyzed databases, high *TSPO* gene expression was associated with a significantly shorter median survival time only in the females and not in males (Figure 2).

### 3.2. Patients Characteristics in CCF and CWRU Datasets

Based on the observations of increased *TSPO* mRNA in GBM patients and an association to OS time in a sex-specific manner, we retrospectively analyzed a large dataset that included 441 GBM patients from CCF. The patient’s characteristics based on the *TSPO* rs6971 polymorphism genotype are detailed in Table 1. In total, we identified 34 patients with the homozygous wild-type A/A genotype, 175 patients with the heterozygous A/G genotype, and 232 patients with the homozygous variant G/G genotype. There was a borderline significant difference (*p* = 0.056) in the mean age at diagnosis between the wild-type A/A genotype (54 years old) and the variant G/G genotype (61 years old) or the heterozygous A/G genotype (61 years old). No significant difference was observed in the ratio of males to females between the three different genotypes, maintaining a higher prevalence in males (67%) than in females (33%) regardless of the genotype. Strikingly, the effect of the *TSPO* rs6971 polymorphism on OS time was statistically significant (*p* < 0.001), showing a median of 13 months for the variant G/G genotype patients or a median of 12 months for the heterozygous A/G genotype patients compared with a median of 25 months for wild-type A/A genotype patients. Furthermore, the *TSPO* rs6971 polymorphism was also significantly associated with the PFS time (*p* = 0.003). The variant G/G and the heterozygous A/G genotypes were associated with worse PFS times (seven and five months respectively) when compared with the wild-type A/A genotype (nine months). This indicates that the presence of only one mutated G allele is associated with a shorter OS and PFS time in GBM patients. Patients were also stratified into those receiving surgery and standard of care treatment, Karnofsky Performance Score (KPS), and tumor recurrence, but no significant association was found with any genotype.

As a validation cohort, we retrospectively analyzed the CWRU dataset that included 131 GBM patients. Table 2 details the patient’s characteristics based on the *TSPO* rs6971 polymorphism genotype. Out of the 131 GBM patients included in CWRU database, we identified 10 patients with the homozygous wild-type A/A genotype, 67 patients with the heterozygous A/G genotype, and 54 patients with the homozygous variant G/G genotype. There was no difference in the mean age at diagnosis between any of the genotypes; however, there was a significant difference (*p* = 0.014) in the ratios of males to females between genotypes. The variant G/G genotype was identified in 22% of females and 78% of males as compared with the heterozygous A/G genotype, which was present in 43% of females and 57% of males, and the wild-type A/A genotype identified in 60% of females and 40% of males. No significant association was found between any genotype and the KPS score, tumor recurrence, vital status, and the treatment received.

### 3.3. High Frequency of TSPO rs6971 Polymorphism in the CCF and CWRU Datasets

Out of the 441 GBM patients in the CCF dataset, 92% carry at least one mutated G allele for the *TSPO* rs6971 polymorphism, with a global minor allele frequency (MAF) of 0.725. This *TSPO* SNP polymorphism exhibit a higher frequency in the GBM population analyzed than the previously reported MAF of 0.3 in Caucasian populations [30]. We also observed that out of the 131 GBM patients in the CWRU dataset, 92% of the GBM patients carry at least one mutated G allele for the *TSPO* rs6971 polymorphism, with a MAF of 0.668. Notably, our cohorts are also mostly Caucasian (frequency >90%). Table 3 shows the comparison between both datasets.

### 3.4. The TSPO rs6971 Polymorphism Is Associated with Worse Overall Survival in GBM Patients Stratified by Biological Sex and Treatment

Based on the initial differences observed, we evaluated the association of the *TSPO* rs6971 polymorphism on the OS in GBM patients from the CCF cohort. In the overall population, a statistically significant difference was observed in the median survival time for the variant G/G genotype patients (14.7 months) and the heterozygous A/G genotype patients (13.6 months), as compared with wild-type A/A genotype patients (25.3 months) (*p* = 0.00034; log-rank test) (Figure 3A). When patients were stratified by the biological sex, the variant G/G genotype (14.3 months) and the heterozygous A/G genotype (13.8 months) were associated with a significantly worse OS time only in males as compared with the wild-type A/A genotype (29.9 months) (Figure 3A). This effect was not observed in females (Figure 3A). We further stratified the patients by treatment, comprising those who underwent surgical resection followed by the standard of care and biological sex. The OS time was significantly worse among those variant G/G genotype patients (20.9 months) and the heterozygous A/G patients (18.2 months) that received surgery and standard of care as compared with the wild-type A/A genotype patients (36.2 months) that also received the same treatment (*p* = 0.019; log-rank test) in the overall population. This association was only significantly different in the male subgroup that received surgery and standard of care treatment, where the variant G/G genotype patients (18.5 months) and the heterozygous A/G patients (18.8 months) exhibited a shorter OS time as compared with the wild-type A/A genotype patients (51.3 months) (*p* = 0.0024) (Figure 3B). Again, this effect was not observed in the females who also received the same therapeutic approach (Figure 3B). Similar trends were observed in the CWRU dataset (Appendix A); however, it was not statistically significant due to the cohort being underpowered (only *N* = 10 of the wild type compared to *N* = 54 of the variant).

In addition, we analyzed a third independent cohort using the TCGA-GBM database and found a borderline significant difference (*p* = 0.08; log-rank test) in the median survival time only in males for the variant G/G genotype patients (12.5 months) and the heterozygous A/G genotype patients (12 months) as compared with the wild-type A/A genotype patients (13.5 months). This effect was not observed in females (Appendix A). There was no significant association of the *TSPO* rs6971 polymorphism with the OS time in the overall population for any of the three genotypes (Appendix A). We could not stratify patients based on the treatment received due to a lack of clinical information in the database.

### 3.5. The TSPO rs6971 Polymorphism Is Associated with Worse Progression-Free Survival in GBM Patients Stratified by Biological Sex and Treatment

We further defined the association of *TSPO* rs6971 polymorphism on the overall recurrence probability or PFS of GBM patients in the CCF cohort. Patients with the variant G/G genotype or the heterozygous A/G genotype for the rs6971 SNP exhibited a significantly worse PFS time in the overall population (8.1 months and 7.1 months respectively) as compared with the wild-type A/A genotype (11.8 months) (*p* = 0.015) (Figure 4A). This difference was even more significant only in males (*p* = 0.0076) as compared with females (*p* = 0.055) (Figure 4A). When patients were also stratified by treatment received, the PFS time was significantly worse in males (*p* = 0.015) among those variant G/G genotype patients or the A/G genotype patients (10.5 months and 8.6 months, respectively) as compared with the wild-type A/A genotype patients (25.7 months) that also received the same treatment (Figure 4B). This association was not significant in the overall population nor in females who also received the same therapeutic approach (Figure 4B). 

In the CWRU dataset, we also observed a similar trend to the CCF dataset (Appendix A) that was not statistically significant due to underpower (only *N* = 10 of the wild type compared to *N* = 54 of the variant). When patients were further stratified by treatment received, the PFS time was worst only in the overall population (*p* = 0.018) among those variant G/G genotype patients that received surgery and standard of care (5.8 months) as compared with the wild-type A/A genotype patients (8.5 months) or the heterozygous A/G genotype patients (11.1 months) that also received the same treatment (Appendix A). This significant difference was not observed when patients were stratified by biological sex. We could not analyze the PFS time in the TCGA database due to the lack of clinical information in the database.

## 4. Discussion

In this study, global *TSPO* gene expression was analyzed in publicly available, independent datasets where we observed a statistically significant worse survival time only in female GBM patients with high *TSPO* gene expression. These results showed an association of *TSPO* with biological sex-specific mechanisms in GBM clinical outcomes. This prompted us to perform genotyping of the most frequent *TSPO* polymorphism, rs6971, in GBM patient samples stratified by sex, using a large cohort from the Cleveland Clinic. We showed statistically significantly worse survival time in male carriers of at least one variant G allele in the *TSPO* rs6971 SNP as compared with the carriers of the A/A wild-type genotype. In addition, the rs6971 G/G variant genotype and A/G heterozygous genotype carriers also showed a significantly worse recurrence-free survival time among male GBM patients as compared with the carriers of A/A wild-type genotype. This indicates that the presence of only one mutated G allele in the *TSPO* rs6971 polymorphism could be sufficient to predict a shorter survival time in male GBM patients. We also used two independent GBM patient cohorts to validate our results, and although the OS and PFS analysis did not show statistically significant association with the *TSPO* SNP, we observed that the rs6971 wild-type A/A genotype carriers have a better survival time only in GBM male patients as compared with the carriers of at least one G allele, showing a similar trend to the CCF cohort.

The CCF dataset is a carefully curated cohort from a single institution, based on the clinical characteristics, where all the patients completed the cycles of the standard treatment. The CWRU dataset comprised fewer patients compared to the primary cohort (CCF dataset), and among these, only 10 patients exhibited the wild-type A/A genotype. Moreover, these numbers got even smaller when stratifying the patients by biological sex and treatment received, which drives the statistical power downward. The TCGA dataset is an older cohort where the data were collected from multiple institutions all over the world to reach their accrual targets (usually around 500 specimens per cancer type). This dataset is extremely heterogenous and was acquired as part of routine care and not as part of a controlled research study or clinical trial. The clinical survival endpoint analyses in the TCGA dataset are approximate because of the lack of absolute verification of the cause of death and relatively short and uniformly clinical follow-up records. All these differences among the different cohorts analyzed can explain the statistical limitations found in this study for the CWRU and the TCGA-GBM datasets [49]. To the best of our knowledge, this study is the first to evaluate the effect of *TSPO* genetic polymorphisms on mortality among patients with GBM in a sex-specific context. 

In the CCF dataset, the *TSPO* rs6971 SNP reach a frequency of 0.725, which may indicate that in the GBM patient population, the most common allele for the *TSPO* rs6971 SNP is the mutated one and may have an important role in the pathogenesis of GBM. Because steroid molecules are involved in several biological functions, this *TSPO* polymorphism could be associated with the susceptibility to or protection against diseases that have been associated with a decreased or increased production of steroids.

Interestingly, in all the cohorts evaluated, the prevalence of GBM is higher in males than in females, corroborating the reported sex differences in the incidence of GBM [1]. Furthermore, our results in differential survival times among male and female carriers of distinct germline variants in the *TSPO* gene support the role of gender-specific molecular patterns in GBM. Current epidemiological data indicate that the male to female incidence ratio for GBM is 1.6:1 in the United States [1]. Several studies suggest that patient outcomes also differ between males and females in the adult GBM patient population [50,51]. It is evident that biological sex differences are inherent drivers of GBM incidence and survival; therefore, the elucidation of biological sex-specific mechanisms in GBM has the potential to improve patient outcomes and develop more personalized GBM therapies. 

Elevated *TSPO* expression is a hallmark of gliosis (microglia and astrocytes), which has led to the use of *TSPO* as a robust marker for brain injury and neuroinflammation [13,14], using both ex-vivo and in-vivo imaging methods. *TSPO*-PET imaging is increasingly being used in neuro-oncology, especially in glioma patients, and has been shown to be a useful tool for assessing tumor progression at follow-up [21,52]. Moreover, several reports found a positive correlation between tumor grade and *TSPO* expression, with the highest expression observed in GBM [42,43]. However, little is known about the functional role of *TSPO* in GBM pathophysiology. Interestingly, we observed that *TSPO* has significantly different expression levels among the different GBM genetic subtypes, showing the highest expression in the mesenchymal subtype in the four independent datasets analyzed. Mesenchymal tumors are predominantly IDH wild-type [53,54], and this subtype is associated with poor radiation response and worse survival [55], which is in line with our finding that *TSPO* is highly expressed in IDH-wild type subtype and is correlated with worse overall survival. Furthermore, we found higher *TSPO* gene expression in the newly described mitochondrial and glycolytic/plurimetabolic GBM subtypes [56], which is consistent with the mitochondrial associated function of *TSPO*. While the mitochondrial GBM subtype was associated with the most favorable clinical outcome, with a marked vulnerability to inhibitors of oxidative phosphorylation, the glycolytic/plurimetabolic subtype exhibits a poor prognosis and is sustained by concurrent activation of multiple energy-producing pathways. Interestingly, this cluster is also enriched in mesenchymal and immune-related functions. 

*TSPO* is overexpressed in activated microglia and macrophages in the surrounding areas of GBM [45] and could play an immunomodulatory role in the GBM microenvironment. However, more studies are required to assess the specific mechanisms of *TSPO* in the GBM pathophysiology and to specifically associate *TSPO* gene expression with any of these newly described GBM subgroups. Several reports indicate that *TSPO* impacts the cell bioenergetic profile by modulating ATP production, supporting a pro-proliferative role of *TSPO* in GBM [57,58,59,60]. In contrast, other studies support an anti-proliferative role of *TSPO*, showing that *TSPO* knockdown and *TSPO* ligands promote cell proliferation and migration due to a decrease in apoptosis [61,62,63]. Studies indicate that *TSPO* is involved in the regulation of cell death; however, the exact mechanism and to what extent *TSPO* regulates the resistance of GBM cells to apoptosis is not known. This contradictory evidence may be due to context-dependent factors that may influence the role of *TSPO* in GBM. This may include differences in cell lines, species, experimental conditions, and signaling pathways. Moreover, *TSPO* ligand effects are concentration- and tissue-specific [64,65]. It is important to note that these biological studies have lacked either analysis of or incorporation of TSPO polymorphic genotypes in the study design. Based on the results shown here, it may be plausible to conclude that the differences observed between studies may be due to varying influences/presence of *TSPO* polymorphisms in the study samples.

In addition, the large inter-individual variability found in *TSPO*-PET brain signal in several clinical trials may be explained by the presence of the genetic polymorphisms in the *TSPO* gene, which affects the binding affinity patterns among different *TSPO* radioligands [30,40]. While there is some evidence that *TSPO* genetic polymorphisms, specifically the rs6971 SNP, may be associated with certain psychiatric disorders, such as bipolar and panic disorders [37,66], it is not yet known whether *TSPO* polymorphisms may have clinical implications in the detection and treatment of GBM patients. Further studies will be required to examine these effects in greater detail.

Our findings provide evidence that the status of the *TSPO* polymorphic variant may be useful in predicting GBM patient outcomes in a sex-specific manner. This remains to be validated in a larger curated cohort. In an underpowered cohort, we observed the expected trends seen in this study. Unfortunately, due to the rarity of GBM, this cohort was unpowered to fully validate our initial findings in the CCF cohort. Additional in-vitro and in-vivo studies are warranted to fully assess the role of *TSPO* polymorphisms in GBM, with a specific focus on biological sex differences, but these studies are beyond the scope of this paper. Further studies are required to understand the exact mechanisms and signaling pathways involved in the regulation of GBM malignancy by the different *TSPO* genetic variants. There is evidence that steroid hormone receptors are expressed in GBM [67], and the inhibition of such receptors can promote GBM cell death in vivo and in vitro [68,69,70]. It is interesting to postulate that the association of *TSPO* genetic polymorphisms with steroid hormonal pathways may play a role in GBM development and/or progression, particularly in a sex-dependent manner. 

## 5. Conclusions

Taken together, our data provide evidence suggesting that the *TSPO* rs6971 gene variant may be a useful indicator of survival time in GBM patients, with the presence suggesting a poorer survival outlook in male patients. This is the first study that reveals the potential for a biological sex-specific relation between rs6971 *TSPO* polymorphism and GBM clinical outcome. These results, taken together with other studies, provide evidence that *TSPO* may play a significant role in GBM pathogenesis. Moreover, our results suggest that Ala147Thr *TSPO* gene polymorphism has the potential to be useful as a prognostic marker of OS and PFS in GBM patients. Future studies will focus on independent validation of these sex-based results and the potential of *TSPO* rs6971 polymorphism to be used as a prognostic biomarker for GBM.

## Figures and Tables

**Figure 1 cancers-13-04525-f001:**
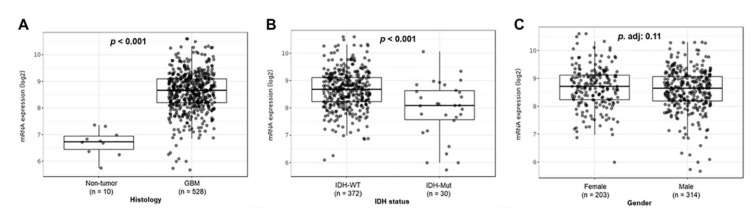
*TSPO* gene-expression analysis in TCGA-GBM cohort. Patient GBM samples from the TCGA-GBM database, using the platform Affymetrix HG-U133A, were analyzed using GlioVis. (**A**) mRNA expression levels in GBM samples compared to tumor-free brain samples. (**B**) Expression levels of *TSPO* in GBM patients grouped by IDH status (IDH-WT vs. IDH-Mut). (**C**) *TSPO* expression levels between males and females with GBM. The number of patients (n) and *p*-value of the pairwise comparisons using *t*-test (with Bonferroni correction) are indicated in each graph.

**Figure 2 cancers-13-04525-f002:**
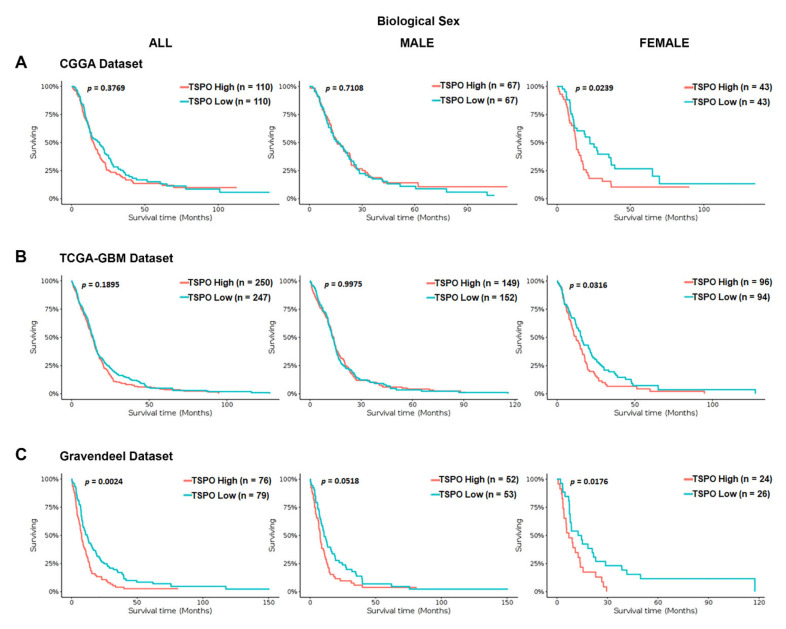
Kaplan–Meier survival curves of *TSPO* mRNA expression in GBM patients stratified by biological sex using the CGGA, TCGA-GBM, and Gravendeel datasets. Overall survival of GBM patients with a high or low expression of *TSPO*, using the median as the cutoff value, was analyzed using the (**A**) CGGA database, (**B**) the TCGA-GBM database (Affymetrix HG-U133A platform), and (**C**) Gravendeel database, available at GlioVis data portal. Survival curves analysis were stratified by biological sex, including the entire population (**left panel**), male population (**middle panel**), and the female population (**right panel**) for the three independent databases. Genotype patient numbers and *p*-values are indicated in each graph, and the median and statistical data are in Appendix A.

**Figure 3 cancers-13-04525-f003:**
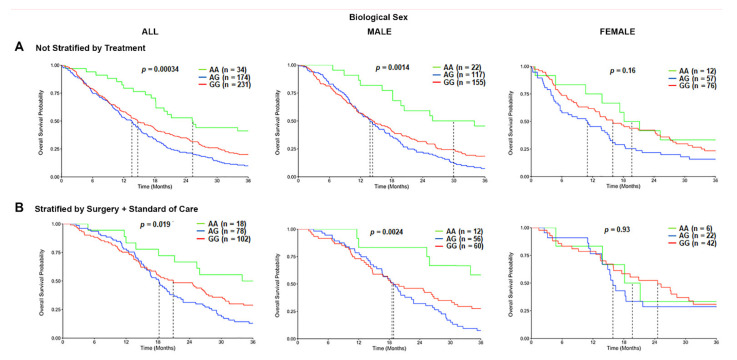
Overall survival (OS) curves in the carriers of the *TSPO* rs6971 polymorphism stratified by genotype, biological sex, and treatment, using the CCF cohort. (**A**) Overall survival probability curves in the entire population (**left panel**), male subgroup (**middle panel**), and female subgroup (**right panel**) comparing the wild-type A/A, heterozygous A/G, and variant G/G genotypes for the *TSPO* rs6971 polymorphism using the patient data from the CCF cohort. (**B**) Overall survival curves stratifying by treatment (those who underwent surgical resection followed by standard of care treatment) in the entire population (**left panel**), the male subgroup (**middle panel**), and the female subgroup (**right panel**) comparing the wild-type A/A, heterozygous A/G, and variant G/G genotypes for the *TSPO* rs6971 polymorphism. Genotype patient numbers and *p*-values are indicated in each graph, and the median and statistical data are in Appendix A.

**Figure 4 cancers-13-04525-f004:**
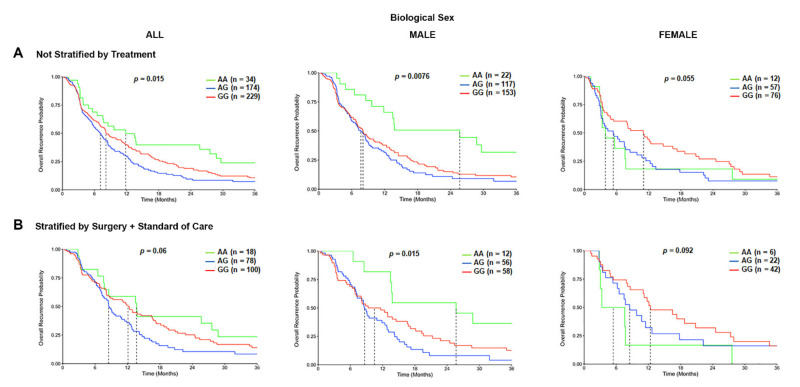
Progression-free survival (PFS) curves in the carriers of *TSPO* rs6971 polymorphism stratified by genotype, biological sex, and treatment, using the CCF cohort. (**A**) Overall recurrence probability curves in the entire population (**left panel**), male subgroup (**middle panel**), and female subgroup (**right panel**) comparing the wild-type A/A, heterozygous A/G, and variant G/G genotypes for the *TSPO* rs6971 polymorphism using patient data from the CCF cohort. (**B**) Overall recurrence probability curves stratifying by treatment (those who underwent surgical resection followed by standard of care treatment) in the entire population (**left panel**), the male subgroup (**middle panel**), and the female subgroup (**right panel**) comparing the wild-type A/A, heterozygous A/G, and variant G/G genotypes for the *TSPO* rs6971 polymorphism. Genotype patient numbers and *p*-values are indicated in each graph, and the median and statistical data are in Table 1.

**Table 1 cancers-13-04525-t001:** Patient Characteristics by *TSPO* rs6971 polymorphism genotype from the CCF dataset.

Characteristics	Overall *N* = 441 ^1^	G/G Variant *N* = 232 ^1^	A/G Heterozygous *N* = 175 ^1^	A/A Wild Type *N* = 34 ^1^	*p*-Value ^2^
Age at Diagnosis	61 (53, 70)	61 (53, 70)	61 (54, 70)	54 (46, 67)	0.056
Sex					>0.9
Female	145 (33%)	76 (33%)	57 (33%)	12 (35%)	
Male	296 (67%)	156 (67%)	118 (67%)	22 (65%)	
Surgery					0.9
Gross Total Resection	152 (49%)	79 (49%)	59 (48%)	14 (48%)	
Near Total Resection	73 (23%)	37 (23%)	31 (25%)	5 (17%)	
Subtotal Resection	88 (28%)	46 (28%)	32 (26%)	10 (34%)	
Unknown	128	70	53	5	
Standard of Care					03
No	66 (15%)	36 (16%)	28 (16%)	2 (5.9%)	
Yes	375 (85%)	196 (84%)	147 (84%)	32 (94%)	
KPS					>0.9
<=70	62 (14%)	32 (14%)	25 (15%)	5 (15%)	
70+	370 (86%)	195 (86%)	147 (85%)	28 (85%)	
Unknown	9	5	3	1	
Overall Survival (Months)	14 (7, 26)	13 (7, 27)	12 (6, 20)	25 (16, 52)	<0.001
Vital Status					0.009
Alive	42 (9.6%)	31 (13%)	8 (4.6%)	3 (8.8%)	
Deceased	397 (90%)	200 (87%)	166 (95%)	31 (91%)	
Unknown	2	1	1	0	
Progression Free Survival	6 (3, 12)	7 (3, 13)	5 (3, 10)	9 (4, 27)	0.003
Recurrence					0.5
No	114 (26%)	60 (26%)	48 (28%)	6 (18%)	
Yes	323 (74%)	169 (74%)	126 (72%)	28 (82%)	
Unknown	4	3	1	0	

^1^ Median (IQR); *n* (%) ^2^ Kruskal–Wallis rank-sum test; Pearson’s chi-square test; Fisher’s exact test. Note: Unknown demographic and clinical characteristics are included in the table but are not assessed in statistical tests.

**Table 2 cancers-13-04525-t002:** Patient Characteristics by *TSPO* rs6971 polymorphism genotype from the CWRU dataset.

Characteristic	Overall *N* = 131 ^1^	G/G Variant *N* = 54 ^1^	A/G Heterozygous *N* = 67 ^1^	A/A Wild Type *N* = 10 ^1^	*p*-Value ^2^
Age at Diagnosis	64 (55, 69)	64 (55, 70)	62 (56, 68)	62 (50, 71)	0.5
Sex					0.014
Female	47 (36%)	12 (22%)	29 (43%)	6 (60%)	
Male	84 (64%)	42 (78%)	38 (57%)	4 (40%)	
Surgery					0.7
Gross Total Resection	75 (59%)	33 (62%)	37 (57%)	5 (50%)	
Subtotal Resection	53 (41%)	20 (38%)	28 (43%)	5 (50%)	
Unknown	3	1	2	0	
Standard of Care					0.7
No	45 (37%)	20 (39%)	21 (33%)	4 (44%)	
Yes	78 (63%)	31 (61%)	42 (67%)	5 (56%)	
Unknown	8	3	4	1	
KPS					0.7
≤70	56 (60%)	23 (61%)	29 (62%)	4 (44%)	
70+	38 (40%)	15 (39%)	18 (38%)	5 (56%)	
Unknown	37	16	20	1	
Overall Survival (Months)	12 (5, 21)	10 (5, 16)	13 (6, 22)	18 (9, 24)	0.4
Vital Status					0.7
Alive	3 (2.3%)	2 (3.7%)	1 (1.5%)	0 (0%)	
Deceased	128 (98%)	52 (96%)	66 (99%)	10 (100%)	
Progression Free Survival	8 (4, 12)	6 (4, 10)	10 (5, 13)	9 (5, 17)	0.11
Unknown	57	22	32	3	
Recurrence					0.6
No	56 (43%)	22 (41%)	31 (47%)	3 (30%)	
Yes	74 (57%)	32 (59%)	35 (53%)	7 (70%)	
Unknown	1	0	1	0	

^1^ Median (IQR); *n* (%) ^2^ Kruskal–Wallis rank-sum test; Fisher’s exact test Note: Unknown demographic and clinical characteristics are included in the table but are not assessed in statistical tests.

**Table 3 cancers-13-04525-t003:** Comparison between CCF and CWRU datasets for *TSPO* rs6971 polymorphism.

Characteristic	Overall *N* = 572 ^1^	CCF Dataset *N* = 411 ^1^	CWRU Dataset *N* = 131 ^1^	*p*-Value ^2^
Age at Diagnosis	62 (53, 70)	61 (53, 70)	64 (55, 69)	0.2
Sex				0.6
Female	197 (34%)	150 (33%)	47 (36%)	
Male	385 (66%)	301 (67%)	84 (64%)	
Race				0.9
Black/African-American		12 (2.8%)	4 (3.1%)	
White		408 (94%)	127 (97%)	
rs6971 SNP Status				0.056
G/G variant	286 (50%)	232 (53%)	54 (41%)	
A/G heterozygous	242 (42%)	175 (40%)	67 (51%)	
A/A wild type	44 (7.7%)	34 (7.7%)	10 (7.6%)	
Unknown	10	10	0	

^1^ Median (IQR); *n* (%). ^2^ Wilcoxon rank-sum test; Pearson’s chi-square test; Fisher’s exact test.

## Data Availability

Publicly available datasets were analyzed in this study. The TCGA dataset can be found here: (https://portal.gdc.cancer.gov/, accessed on 10 May 2021), the CGGA dataset can be found here: (http://www.cgga.org.cn/, accessed on 10 May 2021), The Gravendeel and Rembrandt datasets are available in the GEO database with the accession numbers (GSE16011) and (GSE108474), respectively. The CCF and CWRU datasets presented in this study are available upon request from the senior authors. The data are not publicly available as they are not deidentified.

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
