# Peer review of "The Translocator Protein (TSPO) Genetic Polymorphism A147T Is Associated with Worse Survival in Male Glioblastoma Patients"

_cancers, 2021, doi:10.3390/cancers13184525_

Round 1
Reviewer 1 Report
Overall, this is a very well-written study that evaluates the effect of the rs6971 polymorphism to clinical outcomes of GBM patients while stratifying for various demographic and patient characteristics. Through this approach, the authors suggest an association and highlight the predictive and prognostic value for male patients carrying at least one mutated G allele.
Below are a few comments that should be addressed:
- Tukey’s HSD may not be the most appropriate test for the differential expression analysis of TSPO in GBM patient samples since the groups are of unequal size. Perhaps a pairwise t-test or Tukey-Kramer test should be used.
- The authors calculate the K-M curves for TSPO-High and TSPO-low patients using the median as the cut-off value. It would be informative to try other cut off values (e.g. bottom 25% percentile vs top 25% percentile) and evaluate the association of high TSPO expression and survival.
- As rs6971 is predominant on the Caucasian population, results should also be adjusted or stratified by ethnicity (if data are available). In addition to this, the authors could use a multivariate approach to analyze the survival data while adjusting for key characteristics such as sex, treatment, ethnicity.
- The authors should elaborate more on the connection between the transcriptional findings of the study (higher TSPO expression associated with worse prognosis only in female GBM patients) and the genetic findings (the negative effect of the G allele only in male GBM patients).
- The length of the discussion could be reduced a little bit. Some parts would serve the readers better if they are mentioned in the introduction (e.g. lines 378-400)
Minor comments:
- The R package (and version) that was used for the K-M curves and the log rank test should be mentioned
- Versions of datasets used (or date accessed) should be included
Reviewer 2 Report
This manuscript from Troike and colleagues describes how the translocator protein (TSPO) is significantly overexpressed in glioblastoma (GBM). The TSPO protein has been used as a biomarker for the development of compounds used in neuroimaging of brain tumors. However, the TSPO gene has at least two identified and common variants named rs6971 and rs6972 which significantly hampers the specificity of e.g. radiolabelled probes and induces a strong variability in the neuroimaging readouts.
The authors have used publicly available patient databases coupled to a national dataset and were able to pinpoint that female GBM patient overexpressing TSPO could see their survival very significantly shortened. However, when considering the specific rs6971 polymorphism, the survival trend ended up being less favorable for male GBM patients. This clearly indicates sex differences for brain tumor patients and therefore lays the ground for increased vigilance when interpreting data related to biomarkers.
The study is very thorough and very well conducted and the manuscript very well written.
I have only one comment to be addressed.
Since TSPO is a gene with a strong relationship with the mitochondrion, I would like the authors to at least discuss their results in light of the recent findings from Garofano et al., 2021 (PMID 33681822). Did the authors verify whether TSPO would specifically associate with the newly identified GBM mitonchondrial subtype? As the source data for the patient material sequencing is available, I encourage the authors to, at least, verify a potential association with this subtype. This could possibly emphasize even more the importance of TSPO for GBM progression in male/female patients.
Reviewer 3 Report
This manuscript investigated the association between one single-nucleotide polymorphism (SNP) rs6971 of the translocator protein 18kDa (TSPO) and glioblastoma (GBM) clinical outcomes. The premise of the study is that high TSPO expression correlates with wore survival in GBM, which is most frequent malignant brain tumor in adults. But little is known about the association between SNP rs6971 of TSPO and GBM) clinical outcomes. Using a cohort of 441 GBM patients, the authors found that the TSPO rs6971 polymorphism is associated with worse overall survival in GBM patients stratified by the gender and treatment. Moreover, the same trend was observed in another cohort and there is a borderline significant difference in the median survival time in males from TCGA-GBM database. The authors also found that the TSPO rs6971 polymorphism is associated with worse progression-free survival in GBM patients stratified by the gender and treatment.
This manuscript provided new information to role of the SNP rs6971 in GBM patients, which has the potential to be useful as prognostic marker.
It will be nice to see the analyses using SNP rs6971 together with most common mutations in GBM, such as PTEN, TP53, EGFR, PIK3R1, PIK3CA, NF1, RB1 as the authors have done with IDH1 in Fig 1B.
Reviewer 4 Report
The study by Troike and collaborators analyse TSPO genetic polymorphism A147T on glioblastoma survival and progression-free survival on males and females in three different patient cohorts.
This is an elegant retrospective study that takes advantage of the use of well-curated databases as well as older and highly used ones, like TCGA.
The author reports that this polymorphism is indicative of poor survival and response to treatment in males compared to females and discuss the possible implications of this finding.
I agree with the authors that functional experiments are needed but these are beyond the scope of the present manuscript.
I have only two comments for the authors, that I think can strengthen the conclusions of the manuscript.
The first would be, if it is possible, to include the proportion of GBM subtypes present in the different cohorts (e.g. classical, proneural, mesenchymal) to see whether differences in survival are or not a consequence of dealing with cohorts with different proportions of GBM subtypes.
The second aim is related to the possible functions of TSPO and its polymorphism in GBM, which can be investigated by searching for differential expression genes in patients with low/high TSPO gene expression, and with the different TSPO polymorphisms. Would this analysis be possible (e.g. using TCGA data)?
Round 2
Reviewer 2 Report
I would like to thank the authors for addressing my comment. The additional data analyses and findings are very interesting and relevant to the study. The manuscript is acceptable for publication in my opinion.
Reviewer 3 Report
I Agree to publish.